# A New Scheme of Adaptive Covariance Inflation for Ensemble Filtering Data Assimilation

**Ang Su [1], Liang Zhang [1,\*], Xuefeng Zhang [1,\*], Shaoqing Zhang [2,3,4,5], Zhao Liu [3] , Caili Liu [3] and Anmin Zhang [1]**

[1] School of Marine Science and Technology, Tianjin University, Tianjin 300072, China; su_ang@tju.edu.cn (A.S.); anmin.zhang@tju.edu.cn (A.Z.)

[2] Key Laboratory of Physical Oceanography, MOE, Institute for Advanced Ocean Study, Frontiers Science Center for Deep Ocean Multispheres and Earth System (DOMES), Ocean University of China, Qingdao 266100, China; szhang@ouc.edu.cn

[3] The College of Ocean and Atmosphere, Ocean University of China, Qingdao 266100, China; liuzhaoim@outlook.com (Z.L.); liucaili@stu.ouc.edu.cn (C.L.)

[4] Ocean Dynamics and Climate Function Lab/Pilot National Laboratory for Marine Science and Technology (QNLM), Qingdao 266237, China

[5] International Laboratory for High-Resolution Earth System Prediction (iHESP), Qingdao 266000, China

\* Correspondence: liangzhang@tju.edu.cn (L.Z.); xuefeng.zhang@tju.edu.cn (X.Z.)

**Abstract:** Due to the model and sampling errors of the finite ensemble, the background ensemble spread becomes small and the error covariance is underestimated during filtering for data assimilation. Because of the constraint of computational resources, it is difficult to use a large ensemble size to reduce sampling errors in high-dimensional real atmospheric and ocean models. Here, based on Bayesian theory, we explore a new spatially and temporally varying adaptive covariance inflation algorithm. To increase the statistical presentation of a finite background ensemble, the prior probability of inflation obeys the inverse chi-square distribution, and the likelihood function obeys the *t* distribution, which are used to obtain prior or posterior covariance inflation schemes. Different ensemble sizes are used to compare the assimilation quality with other inflation schemes within both the perfect and biased model frameworks. With two simple coupled models, we examined the performance of the new scheme. The results show that the new inflation scheme performed better than existing schemes in some cases, with more stability and fewer assimilation errors, especially when a small ensemble size was used in the biased model. Due to better computing performance and relaxed demand for computational resources, the new scheme has more potential applications in more comprehensive models for prediction initialization and reanalysis. In a word, the new inflation scheme performs well for a small ensemble size, and it may be more suitable for large-scale models.

**Keywords:** ensemble Kalman filter; covariance inflation; coupled model; sampling and model errors

## 1. Introduction

Data assimilation (DA) incorporates observations into a climate model through background error covariances derived from model dynamics and then produces a continuous time series of climate states [1–3]. In the ensemble Kalman filter (EnKF) [4], covariance inflation [5] is often used to avoid underestimating the background error covariance caused by a finite size of ensembles. It increases the state's uncertainty by expanding its ensemble spread and increases the confidence in the observations.

The covariance inflation scheme is mainly divided into multiplicative [6,7], additive [8] and observation error variance [9] inflation. This paper focuses on multiplicative inflation, which is further divided into prior and posterior inflation by applying the inflation factor to the background ensemble and the analysis ensemble, respectively. Prior inflation was proposed earlier, and Anderson [10] used it in the ensemble adjusted Kalman filter (EAKF) assimilation method. The inflation factor here requires manual tuning for each assimilation. Consequently, it is often time consuming and computationally expensive,

especially for complex geophysical models. Many studies have also pointed out that the EnKF assimilation method is sensitive to the choice of the inflation factor [11,12]. Therefore, Anderson [13] developed a time-adaptive covariance inflation algorithm based on hierarchical Bayesian estimation theory. He updated the inflation factor like a variable and got results as satisfactory as those from manual tuning. By extending the Bayesian approach, Anderson [14] proposed a spatial and temporal varying adaptive covariance inflation algorithm in 2009 (A09). In addition, an online inflation factor estimation algorithm in the ensemble transform Kalman filter (ETKF) framework was proposed by Wang et al. [15] and extended by Li et al. [16] to simultaneously estimate covariance inflation and observation errors online. Miyoshi [17] improved the ETKF framework by adaptively estimating the inflation factor at each grid point, and the method has been applied to several geophysical system studies [18–21]. Zheng [22] and Liang [9] used the maximum likelihood method to estimate the inflation factor from the update vector at each time step. Zhang [23] proposed a special posterior inflation (relaxation) scheme, the relaxation-to-prior-perturbation (RTPP) approach. Based on it, Whitaker and Hamill [12] proposed the relaxation-to-prior-spread (RTPS) method. The relaxation factors of these two methods are obtained by manual tuning. Ying and Zhang [24] proposed an adaptive RTPS method, and Kotsuki et al. [25] proposed an adaptive RTPP method to obtain varying optimal factors. Both methods are based on the innovation statistics [8,26] in the observation space.

All the above methods assume that the inflation innovation is Gaussian, but it can produce negative or minimal inflation values, and a long run of deflation may lead to filter divergence. So, scholars have tried many other schemes. For example, Brankart [27] made the initial prior obey the exponential distribution, but it is not suitable for small values. As a conjugate distribution to the variance parameter of the Gaussian distribution, the inverse chi-square ($\chi^{-2}$) (equivalent to the inverse-gamma) distribution may be a better choice. El Gharamti made the prior probabilities of inflation obey the inverse-gamma distribution [28] (E18) and applied it to the posterior inflation [29] (E19). Raanes [30] made the likelihood function obey the $\chi^2$ distribution, and the prior and posterior probabilities obey the $\chi^{-2}$ distribution. However, all these advantages seem insufficient for small ensemble sizes.

In this paper, a new inflation scheme is proposed in the framework of Bayesian theory, in which the prior probability still obeys the $\chi^{-2}$ distribution and the likelihood function obeys the *t* distribution, which is more suitable for small sample sizes, and data assimilation experiments are performed in two atmospheric-ocean-coupled model frameworks. In the results of the comparison, the new scheme shows significant effects under high sampling and model errors.

In addition to the explanation of abbreviations in this paper, we also list main abbreviations and definitions in the text in Appendix A to read more conveniently. The abbreviations of inflation methods are explained as follows: AIb denotes the spatial and temporal adaptive prior covariance inflation scheme in A09; AIa denotes the adaptive inflation scheme that uses AIb into posterior inflation; EIb denotes the enhanced adaptive prior inflation scheme in E18; EIa denotes the enhanced adaptive posterior inflation scheme mAI-a in E19; tXb and tXa are the adaptive prior and posterior inflation schemes proposed in this paper, respectively.

The paper is organized as follows: Section 2 introduces the assimilation method, the basic theory of adaptive inflation and a new adaptive inflation scheme. Section 3 focuses on a series of numerical experiments with a simple five-variable model and compares the new method's performance with other inflation schemes. Section 4 verifies the applicability and effectiveness of the new inflation scheme in another coupled model. Finally, the discussion and conclusion are given in Section 5.

## 2. Methodology

### 2.1. EAKF Assimilation Method

To construct the assimilation frame, we used the ensemble adjustment Kalman filter (EAKF) [3,10,31–33] assimilation method. The process involves two steps. First, the observation increment $\Delta y_{o,i}$ is calculated from the state ensemble $x$ and the observation $y_o$:

$$\Delta y_{o,i} = \frac{\overline{y}}{1 + r^2(y, y_o)} + \frac{y_o}{1 + r^{-2}(y, y_o)} + \frac{y_i - \overline{y}}{\sqrt{1 + r^2(y, y_o)}} - y_i \tag{1}$$

where $y$ is the projection of the state values on the observation space, $y = h(x)$, $h$ is the projection operator; $y_i$ is the $i$-th member of the ensemble; $\overline{y}$ is the ensemble mean; and $r^2(y, y_o)$ is the ratio of the model ensemble variance in the observation space and the observation error variance, i.e., $\sigma_y^2 / \sigma_{y_o}^2$.

Second, the state increment $\Delta x_i$ is calculated from the observation increment:

$$\Delta x_i = \frac{cov(x, y)}{\sigma_y^2} \Delta y_{o,i} \tag{2}$$

where $cov(x, y)$ is the error covariance of the state ensemble $x$ and the ensemble $y$ in the observation space. The assimilated analysis ensemble is obtained by adding the state increments to each corresponding member of the state ensemble.

### 2.2. Adaptive Inflation Algorithm

#### 2.2.1. Basic Inflation Theory

This section focuses on the basic inflation scheme to be compared in this paper, and all the equations can be found in A09, E18 and E19. To compensate for the error covariance lost in the ensemble assimilation process and prevent filter divergence, the error covariance needs to be inflated, i.e., $P_{inf} = \lambda P$, where $\lambda$ is the inflation factor, which is generally slightly greater than 1. In practice, the inflation of error covariance is generally achieved by inflating the state ensemble spread, as shown in Equation (3). For simplicity, we assume that all equations are at the same time step, so the time subscripts are omitted:

$$x_{j,i}^{inf} = \sqrt{\lambda}\left(x_{j,i} - \overline{x}_j\right) + \overline{x}_j \tag{3}$$

where $x_{j,i}$ denotes the $i$-th member state value of the $j$-th variable and $\overline{x}_j$ denotes the ensemble mean of the $j$-th variable. A larger ensemble size corresponds to a smaller inflation factor and vice versa [6]. In the background state framework, considering only scalar systems (which can be extended to vector systems), the background ensemble mean $\overline{x}_b$ and sample variance $\hat{\sigma}_b^2$ are expressed as follows:

$$\overline{x}_b = \frac{1}{N} \sum_{i=1}^{N} x_{b,i} \tag{4}$$

$$\hat{\sigma}_b^2 = \frac{1}{N-1} \sum_{i=1}^{N} (x_{b,i} - \overline{x}_b)^2 = \frac{1}{N-1} \sum_{i=1}^{N} {x'_{b,i}}^2 \tag{5}$$

where $b$ (background) denotes the prior inflation. This can be replaced by $a$ (analysis) denoting the posterior inflation, to represent the statistical analysis of the posterior ensemble. $x'_{b,i}$ denotes the ensemble perturbation of the background ensemble, and $N$ is the ensemble size. Theoretically, the background ensemble member $x_{b,i} \sim N(\mu, \sigma_b^2)$, but the background variance $\hat{\sigma}_b^2$ calculated from the sample is an underestimated variance due to reasons such as the finite ensembles:

$$\sigma_b^2 = \lambda \hat{\sigma}_b^2 \tag{6}$$

That is, the underestimated variance is inflated to obtain the true or near-true variance. Assuming that the true state of the model is $x_t$, the observation $y_o$ can be obtained by the following equation:

$$y_o = h(x_t) + \varepsilon_o \tag{7}$$

where the operator $h$ is used to project the state space variables into the observation space (both are consistent by default in this paper, so $h$ is the unit matrix and is omitted after here). $\varepsilon_o$ is the observation error, which is set to obey a Gaussian distribution with mean 0 and variance $\sigma_o^2$. Similarly, we can obtain:

$$\overline{x_b} = x_t + \varepsilon_b \tag{8}$$

$$\overline{x_a} = x_t + \varepsilon_a \tag{9}$$

where the background error $\varepsilon_b$ and the analysis error $\varepsilon_a$ obey a Gaussian distribution with mean 0 and variance $\sigma_b^2$, $\sigma_a^2$, respectively. The analysis ensemble is calculated from the background ensemble and the observation correction as follows:

$$x_{a,i} = x_{b,i} + f(x_{b,i}, y_o) \tag{10}$$

where the correction term $f(x_{b,i}, y_o)$ is a function of $x_{b,i}, y_o$.

According to the definition of the innovation statistic, the background distance $d_b$ is given by Equation (11):

$$d_b = y_o - \overline{x_b} = \varepsilon_o + x_t - \overline{x_b} = \varepsilon_o - \varepsilon_b \tag{11}$$

The innovation statistics respond to the difference between the observation and the ensemble mean, which is used later in Bayesian theory to calculate the likelihood of inflation. In addition, since the background ensemble is formed by adding random perturbations directly to the initial field, we can assume that the background error $\varepsilon_b$ is not related to the observation error $\varepsilon_o$. However, the analysis ensemble is calculated from the background ensemble and observations, so the analysis error $\varepsilon_a$ is considered related to the observation error $\varepsilon_o$.

With the development of inflation theory, prior inflation has been widely studied and applied, as described in Section 1. It inflates the prior ensemble of states using Equation (3) before the assimilation step. A09 proposed a classical spatial–temporal adaptive inflation algorithm, which is called AI-b in this paper. Similar to the estimation of state variables, the inflation factor as a parameter also requires prior inflation and observations through Bayesian theory to compute the posterior inflation:

$$p(\lambda|d_b) = p(d_b\lambda)p(\lambda)/norm \propto p(d_b\lambda)p(\lambda) \tag{12}$$

where $p(\lambda|d_b)$ is the posterior probability of $\lambda$. Equation (12) is used to calculate the value of $\lambda$ when the posterior probability is the maximum. $p(\lambda)$ is the prior probability of $\lambda$ with a model function of 1, i.e., the posterior inflation factor is the prior factor the next time. Anderson considered the prior probability to obey a Gaussian distribution with mean $\overline{\lambda}_b$ and variance $\sigma_{\lambda,b}^2$. Here, *norm* is a standardized constant. $p(d_b|\lambda)$ is the likelihood, which is also considered to obey a Gaussian distribution, and its mean and variance of the background innovation statistic on the prior $\lambda$ are given by the following equations, respectively:

$$E(d_b) = E(\varepsilon_o - \varepsilon_b) = 0 \tag{13}$$

$$D(d_b) = E\left(d_b{}^2\right) = E\left(\varepsilon_o{}^2 + \varepsilon_b{}^2 - 2\varepsilon_o\varepsilon_b\right) = \sigma_o^2 + \sigma_b^2 \tag{14}$$

Since $\varepsilon_o$ is not related to $\varepsilon_b$, the expectation of its product is 0. The variance of the likelihood is denoted as $\theta^2 = \sigma_o^2 + \lambda\hat{\sigma}_b^2$ by matching Equation (6) and a determined

observation error variance. Further, the inflation posterior probability density function (pdf) can be obtained as follows:

$$p(\lambda|d_b) = \left(\sqrt{2\pi}\theta\right)^{-1} e^{-\frac{d_b^2}{2\theta^2}} \cdot \left(\sqrt{2\pi}\sigma_{\lambda,b}\right)^{-1} e^{-\frac{(\lambda_b - \bar{\lambda}_b)^2}{2\sigma_{\lambda,b}^2}} \tag{15}$$

To calculate $\lambda$ when the posterior probability is maximized, the above equation is derived, and its final form is for a cubic equation [13]. However, if there is non-exact correspondence between the observation space and the state space, the influence of the correlation coefficient between the observation and the prior state or localization factor should be considered [14]. The inflation factor in the observation space should be a function related to it in the state space:

$$\lambda_o = \left[1 + \gamma\left(\sqrt{\lambda_b} - 1\right)\right]^2 \tag{16}$$

where $\gamma = \rho r$, $\rho$ is the localization factor, and $r$ is the correlation coefficient between observation and state. If $\gamma \neq 1$, the result based on Equation (15) is a sixth-order equation, which is generally insoluble. Thus, based on Equation (15), A09 performed a Taylor expansion on the likelihood function, the linear term was retained, and finally a quadratic equation concerning $\lambda$ was obtained, giving a solution close to $\lambda_b$.

Ideally, $\lambda$ should be greater than 1 to push the ensemble state away from its mean and increase the error covariance. When $\lambda = 1$, no inflation is performed. If $\theta^2 - \sigma_o^2 < \hat{\sigma}_b^2$ or $\theta^2 < \sigma_o^2$, $\lambda$ will be less than 1 or even less than 0, which will not inflate. Similarly, $\lambda$ much larger than 1 is also infeasible, which would lead to over-inflation of the ensemble. Therefore, $\lambda$ should be a reasonable range of variation.

Unlike A09, in the enhanced prior scheme (EIb) in E18, El Gharamti got the innovation statistic $d$ in the likelihood function by each member of the prior ensemble:

$$d_{b,i} = y_o - x_{b,i} \tag{17}$$

By calculation, the expectation of $d$ remains the same and the variance is added to the original with a correction term related to the ensemble size. The modified variance of the likelihood function is:

$$\theta^2 = \sigma_o^2 + \left\{\left[1 + \gamma\left(\sqrt{\lambda_b} - 1\right)\right]^2 - \frac{1}{N}\right\}\hat{\sigma}_b^2 \tag{18}$$

Meanwhile, the prior probability in E18 obey the inverse-gamma distribution, as shown in the following equation:

$$p(\lambda) = \frac{\beta^\alpha}{\Gamma(\alpha)}\lambda^{-\alpha-1}\exp\left(-\frac{\beta}{\lambda}\right) \tag{19}$$

where $\alpha$ is the shape parameter, $\beta$ is the rate parameter, $\Gamma$ is the gamma function and $\Gamma(x) = \int_0^{+\infty} t^{x-1}e^{-t}dt$, $(x > 0)$. If the prior mean (mode) and variance of the Gaussian distribution are available, the two unknown parameters can be found by making them equal to the mode and variance of the inverse-gamma distribution, respectively. Compared with the Gaussian distribution, the inverse-gamma distribution features of this scheme avoid some negative or small inflation values and reduce the impact on the assimilation quality.

Different from the prior inflation, in the enhanced posterior inflation scheme (EIa) in E19, in addition to the inflation factor is acting on the analysis ensemble, the following treatment is applied to the analysis state and variance.

$\varepsilon_o$ is related to $\varepsilon_a$, the likelihood variance is not the same as the results of Equations (14) and (18), but a function of the posterior variance $\sigma_{a,j}^2$, the posterior variance of the previous assimilation step $\sigma_{a,j-1}^2$ and the observation variance $\sigma_{o,j}^2$, where $j$

denotes the *j*-th observation of the assimilation [29]. To reduce the computational cost in the high-dimensional complex model, El Gharamti decorrelated them as follows:

$$\widetilde{\sigma}_a^2 = \frac{1}{\frac{1}{\sigma_a^2} - \frac{1}{\sigma_o^2}} \tag{20}$$

$$\widetilde{x}_a = \widetilde{\sigma}_a^2 \left( \frac{x_a}{\sigma_a^2} - \frac{y_o}{\sigma_o^2} \right) \tag{21}$$

where $\widetilde{\sigma}_a^2$ and $\widetilde{x}_a$ are not correlated with observations and the innovation statistic becomes $d_{a,i} = y_o - \widetilde{x}_{a,i}$. To ensure that the variance $\widetilde{\sigma}_a^2$ is not less than 0, several methods are designed to restrict it. A comparison of the experimental effects shows that $\widetilde{\sigma}_a^2 = \sigma_a^2$ when $\sigma_a^2 > \sigma_o^2$.

### 2.2.2. The New Inflation Scheme

The classical adaptive inflation is computed based on Bayesian theory with a Gaussian framework. However, as described in Section 1, many studies have shown that a framework with a Gaussian distribution is not the only choice and different distributions have some advantages in some aspects. In this paper, we used alternative distributions to obtain new inflation schemes.

#### Prior Probability

Raanes showed that the inverse-gamma or inverse chi-square distribution is a better choice for the prior pdf of inflation, which is also better than the assimilation effect in the Gaussian framework (note that the gamma and chi-square distributions are equivalent and can be converted into each other [30]). Therefore, this scheme uses the inverse-gamma distribution as in E18 to describe the prior pdf of inflation, as in Equation (19).

#### Likelihood Function

When the degree of freedom is large enough, the *t* distribution is believed to become the same as the Gaussian distribution. However, when the sample size is small, the *t* distribution shows the feature of "heavy-tailed." It is influenced by the sample and deviates significantly from the Gaussian distribution. Large ensemble sizes cannot be used in actual large-scale climate models, so the *t* distribution is more suitable than the Gaussian distribution for estimating the overall population. For the above reasons, the inflation scheme makes the likelihood function obey the *t* distribution, where the pdf of the t-distribution is derived from Table A1 in the paper by Raanes [30]:

$$p(d|\lambda) = c_t |B|^{-\frac{1}{2}} \left( 1 + \frac{1}{v} \frac{(d-b)^2}{B} \right)^{-\frac{v+M}{2}} , \; c_t = \frac{\Gamma\left(\frac{v+M}{2}\right)}{(\pi v)^{\frac{M}{2}} \Gamma\left(\frac{v}{2}\right)} \tag{22}$$

where $v$ is the degree of freedom, which is equal to the ensemble size; $M$ is the number of state variables; and $b$ and $B$ are the parameters in the *t* distribution pdf. The *t* distribution has mean $b$ and variance $v/(v-2)B$. To find the required parameters, suppose the prior mean and variance of the Gaussian distribution are available so that both means and variances are equal:

$$b = 0, \quad \frac{v}{v-2} B = \theta^2 \tag{23}$$

Bringing them into Equation (22), the likelihood function can be obtained as follows:

$$p(d|\lambda) = c_t \left| \frac{v-2}{v} \theta^2 \right|^{-\frac{1}{2}} \left[ 1 + \frac{d^2}{(v-2)\theta^2} \right]^{-\frac{v+M}{2}} , \; c_t = \frac{\Gamma\left(\frac{v+M}{2}\right)}{(\pi v)^{\frac{M}{2}} \Gamma\left(\frac{v}{2}\right)} \tag{24}$$

To enhance the relationship between the innovation statistic $d$ and each state ensemble member, the method proposed in E18 can be used. So, the variance of the likelihood function in the scheme is shown as Equation (18).

Posterior Probability

According to Bayesian theory (Equation (12)), multiplying the likelihood function with the prior probability gives the posterior probability of inflation:

$$p(\lambda|d) = c_t \left| \frac{v-2}{v} \theta^2 \right|^{-\frac{1}{2}} \left[ 1 + \frac{d^2}{(v-2)\theta^2} \right]^{-\frac{v+M}{2}} \frac{\beta^\alpha}{\Gamma(\alpha)} \lambda^{-\alpha-1} \exp\left( -\frac{\beta}{\lambda} \right) \qquad (25)$$

where $\theta$ is a function of $\lambda$ and $\alpha$ and $\beta$ are functions of $\lambda_b$ and $\sigma_\lambda^2$. Therefore, finding the updated posterior inflation is equivalent to finding the value of $\lambda$ when the posterior probability is maximized.

Let the derivative of the posterior probability be 0. Eventually, the same quadratic equation as Equation (38) in E18 [28] can be obtained:

$$\left( 1 - \frac{\lambda_b}{\beta} \right) \lambda^2 + \left( \frac{\bar{l}}{l'} - 2\lambda_b \right) \lambda + \left( \lambda_b^2 - \frac{\bar{l}}{l'} \lambda_b \right) = 0 \qquad (26)$$

However, $\bar{l}$ and $l'$ are not same with them in E18, the detailed procedure can be found in Appendix B. The root close to $\lambda_b$ is the updated inflation factor. In the posterior inflation scheme, we also use the scheme in E19 for decorrelation, as shown in the previous section.

Since the inflation method is obtained from the likelihood function obeying the $t$ distribution and the prior probability obeying the $\chi^{-2}$ distribution, they can be used for the background state to obtain the prior inflation (tXb) and the analysis state to obtain the posterior inflation (tXa), respectively. The updated inflation variance calculation is not given here because a fixed variance is more appropriate [13] in terms of the calculation's cost and effectiveness. It is proved that even the adaptive varying covariance decreases to a stable value over time [28].

Algorithm Implementation

The computing process and characteristics of the new adaptive inflation algorithm in the sea-air coupled assimilation model based on EAKF are as follows:

- Without abandoning the Gaussian framework, the $t$ distribution of the likelihood function and the $\chi^{-2}$ distribution of the prior probabilities are used, assuming that their Gaussian distributions are available, and their product outputs are Gaussian priors when assimilating the next observation.
- The prior inflation factor is used before each variable assimilation step, and the posterior inflation factor is used after it.
- Localization is not considered in this paper, and since the state space and the observation space are consistent, $\gamma = 1$.
- The rate parameter $\beta$ is calculated by the mean $\lambda_b$ and variance $\sigma_{\lambda,b}^2$ of the prior inflation factor.
- The innovation statistic $d$ as well as its variance $\bar{\theta}^2$ are calculated. Then, the ratio of the gamma function is calculated by the special method proposed in this paper to obtain the values of $\bar{l}$ and $l'$.
- Finally, the quadratic equation containing $\beta$, $\bar{l}$, $l'$ and $\lambda_b$ is solved to obtain the updated inflation factor $\lambda_u$. The new $\lambda_{u,j}$ is the prior inflation factor $\lambda_{b,j+1}$ when assimilating the next observation.

## 3. 5VCCM Experiments

### 3.1. The Model

We first used a five-variable coupled climate model (5VCCM), a decadal pycnocline prediction model, proposed by Zhang [34,35] and widely used in many studies [3,36], to conduct a series of experiments and analyze the experimental results. The 5VCCM is a simple version of the coupled general circulation model (CGCM), with some similar features, avoiding the enormous costs of using complex models. The 5VCCM consists of five variables: three variables from the Lorenz63 chaotic atmosphere model [37], one variable from the slab ocean model, and one variable from the deep-ocean pycnocline model [38]. The fast atmosphere drives the slower ocean, resulting in sea-air interactions. The governing equations are as follows:

$$
\begin{aligned}
\dot{x}_1 &= -\sigma x_1 + \sigma x_2 \\
\dot{x}_2 &= -x_1 x_3 + (1 + c_1 \omega) \kappa x_1 - x_2 \\
\dot{x}_3 &= x_1 x_2 - b x_3 \\
O_m \dot{\omega} &= c_2 x_2 + c_3 \eta + c_4 \omega \eta - O_d \omega + S_m + S_s \cos\left(2\pi t / S_{pd}\right) \\
\Gamma \dot{\eta} &= c_5 \omega + c_6 \omega \eta - O_d \eta
\end{aligned}
\tag{27}
$$

where all quantities are given in non-dimensional units. $x_1$, $x_2$ and $x_3$ are atmospheric variables, where $x_1$ is the flip rate of convection, $x_2$ is the temperature difference proportional between the up-flow and down-flow fluids and $x_3$ is the temperature gradient in the vertical direction. $\omega$ and $\eta$ are ocean variables, where $\omega$ denotes the slab-ocean and $\eta$ denotes the deep-ocean pycnocline. A dot above a variable denotes the time tendency. The above five formulas constitute a system of nonlinear differential equations and contain 15 parameters. $\sigma$, $\kappa$ and $b$ are the original parameters in the Lorenz63 model with standard values of 9.95, 28 and 8/3, respectively. $c_1$ denotes the parameter of atmospheric forcing by the ocean; $c_2$ denotes the atmospheric forcing on the upper ocean; $c_3$ and $c_4$ denote the linear forcing by the deep ocean on the upper ocean and the interaction between them, respectively; and $c_5$ and $c_6$ denote the linear forcing by the upper ocean on the deep ocean and their interaction, respectively. Without the interaction between different media, the upper ocean would consist of only the damping term $O_d \omega$ and the external forcing $S(t) = S_m + S_s \cos\left(2\pi t / S_{pd}\right)$, where $O_d$ is the damping coefficient; $S_m$ and $S_s$ define the magnitude of the annual mean and seasonal cycle, respectively, and $S_{pd}$ defines the timescale of the seasonal cycle. Since the timescale of $\omega$ is much slower than that of the atmosphere, the heat capacity $O_m$ is much larger than the damping coefficient $O_d$, which means that the timescale of the ocean is $O_m/O_d$ times that of the atmosphere. In the deep-ocean pycnocline model, $\eta$ denotes the anomaly of the ocean pycnocline depth, and its equation is derived from the two-term balance model of the zonal-time mean pycnocline [38]; $\Gamma$ is the constant of proportionality.

Following Zhang [34] on the set of parameters, the values of 15 parameters were set in this paper as $\left(\sigma, \kappa, b, c_1, c_2, c_3, c_4, c_5, c_6, O_m, O_d, S_m, S_s, S_{pd}, \Gamma\right) =$ $\left(9.95, 28, 8/3, 10^{-1}, 1, 10^{-2}, 10^{-2}, 1, 10^{-2}, 10, 1, 10, 1, 10, 100\right)$, where $\sigma$, $\kappa$ and $b$ are still selected as the standard values in the Lorenz63 model.

### 3.2. Experiment Design

The experiments were designed to compare the performances of new inflation schemes with those of other inflation schemes with different ensemble sizes. Before starting the assimilation experiments, we needed to construct perfect and imperfect assimilation models. We selected the leapfrog time difference scheme as the perfect model scheme [34] and used the Robert-Asselin time filter [39,40] with a time filter coefficient of 0.125. The fourth-order Runge-Kutta (RK4) time difference scheme was used in the imperfect model for comparison [36]. The experimental time step was $\Delta t = 0.01$, and all 15 parameters were considered as standard values in Section 3.1. We assumed that the only source of the model

error is from the different time difference schemes. The initial values and the observation of the experiment are generated with reference [36], and the true and observation field required for the experiment can be obtained from the initial values and the model of the input parameters together, as follows.

The five variables $(x_1, x_2, x_3, \omega, \eta)$ of the coupled model were spun up from the initial values $(0, 1, 0, 0, 0)$ for 1000 time units (TUs; 1 TU is 100 time steps) in the perfect and imperfect models, respectively, to obtain true and biased initial fields. Then running the true initial field for 10,000 TUs using the perfect model, we obtained the true values of the five variables about the time series. The observation field was obtained by adding Gaussian white noise with a standard deviation of 2 every 5 steps for $x_1, x_2$ and $x_3$, and a standard deviation of 0.2 every 20 steps for $\omega$. This observation frequency was based on the actual climate observation system, where the atmosphere has a higher frequency of observations than the ocean. The deep-ocean variable $\eta$ had no observations, so no inflation was performed on it.

The initial ensembles of perfect and imperfect model assimilations were obtained from true and biased initial fields, respectively, adding only Gaussian white noise consistent with the observed standard deviation on $x_2$. The ensembles were used as the initial condition to run 10,000 TUs corresponding to their time difference methods, respectively, and different inflation schemes were used for comparison. The assimilation effect was judged by the root-mean-square error (RMSE). The RMSE (Equation (28)) time series of $x_2$ in the last 100 TUs, $\omega$ in the last 1000 TUs, and $\eta$ in the 10,000 TUs were selected for analyses and comparison. The mean RMSE (Equation (29)) of the stable last 5000 TUs was also represented.

$$\text{RMSE}_1 = \sqrt{(\overline{x} - x_t)^2} \tag{28}$$

$$\text{RMSE}_2 = \sqrt{\frac{1}{n} \sum_{i=1}^{n} (\overline{x}_i - x_{t,i})^2} \tag{29}$$

where the subscript denoting the state is omitted, $\overline{x}$ is the mean of the state ensemble, and $n$ is the number of steps for analysis.

The initial inflation factor was 1.0. The standard deviation of the inflation factor took a fixed value such that $\sigma_{\lambda_b} = 0.1$ when using the perfect model and $\sigma_{\lambda_b} = 1.0$ when using the imperfect model [14].

The other two experiments were conducted as a reference to the assimilation results. The first was a control (CTRL) experiment that did not introduce any observations, i.e., only model integration was performed. The second was an assimilation experiment with state estimation only (SEO), without introducing covariance inflation or localization.

*3.3. Result Analysis*

Based on the above experimental setup, this section compares and analyzes the performance of the new inflation scheme used in the prior and posterior ensembles in the perfect and imperfect models and shows the effect of the traditional assimilation method with the new adaptive inflation scheme.

3.3.1. Inflation Scheme Comparison
Imperfect Model

- Prior inflation scheme

The initial bias ensemble was integrated using the RK4 difference method with different inflation schemes. The time series of RMSEs compared with SEO and CTRL are shown in Figure 1. The manually tuned inflation factor was almost unavailable in the complex model, so it was not compared in this paper. The black line is the control experiment CTRL, the magenta line is the SEO, the red line is the spatial-temporal adaptive prior inflation method proposed by A09, the green line is the enhanced adaptive prior inflation method proposed by

E18, and the blue line is the new prior inflation scheme proposed. The inset in each graph in Figure 1 shows its partial enlargement for a clear comparison of all inflation schemes.

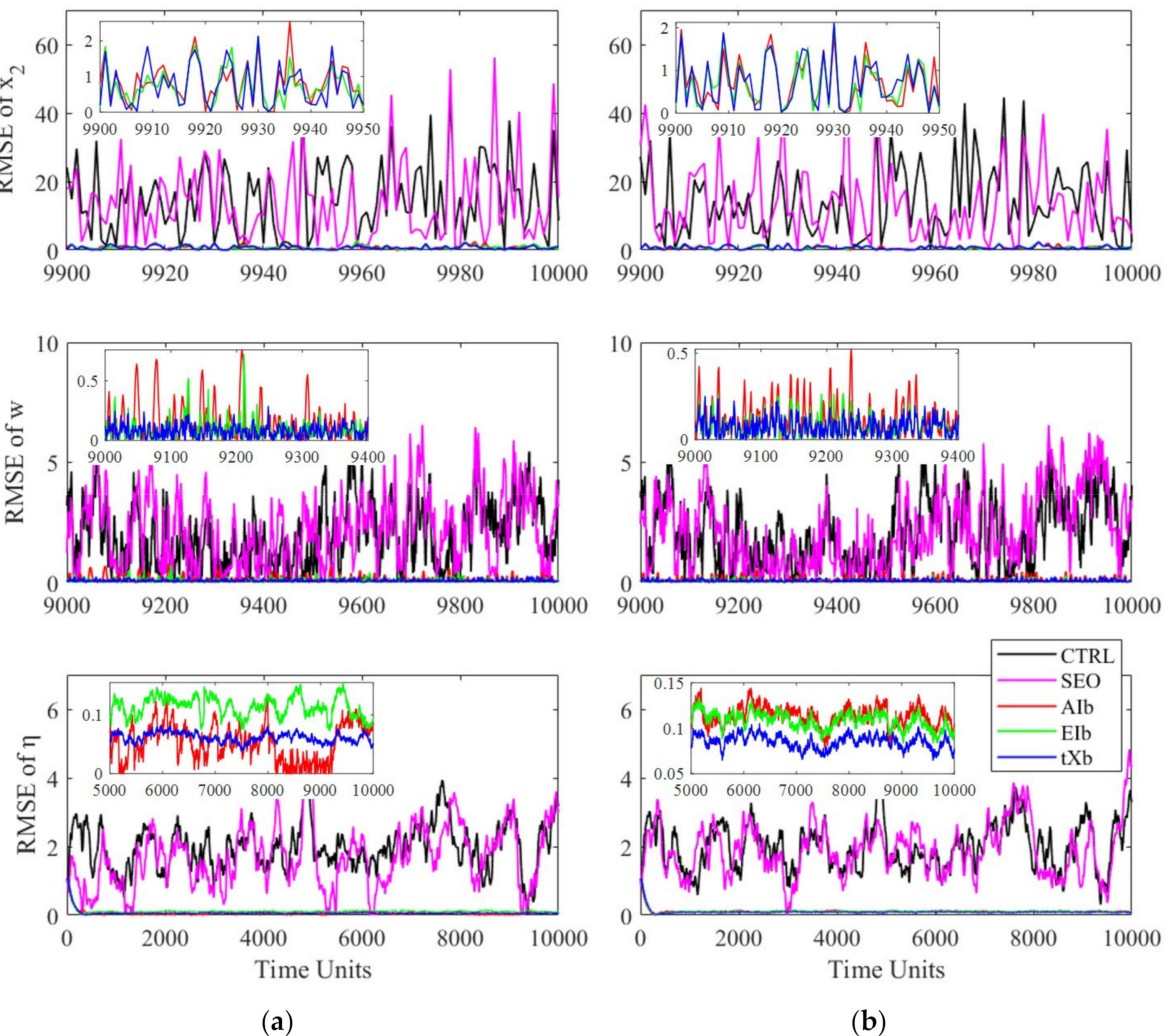

**Figure 1.** Time series of RMSEs of the atmospheric variable $x_2$ in the last 100 TUs, the upper-sea variable $\omega$ in the last 1000 TUs, and the deep-sea variable $\eta$ in 10,000 TUs in the imperfect model with CTRL (black line), SEO (magenta line), AIb (red line), EIb (green line) and tXb (blue line), with an ensemble size of 5 for (**a**) and 20 for (**b**). The inset in each graph shows its partial enlargement to show the time series of RMSEs of $x_2$ in 50 TUs, $\omega$ in 400 TUs and $\eta$ in the last 5000 TUs.

Figure 1a shows the experimental results for an ensemble size of 5. The RMSEs of CTRL and SEO were much larger than those on adding the inflation scheme. In addition, we compared the effects with the three inflation schemes. There was no apparent difference between the three schemes for $x_2$. For the variable $\omega$, AIb performed poorly and often had RMSEs above the observational standard deviation. Both EIb and tXb worked better due to the $\chi^{-2}$ distribution of the inflation prior probability. tXb was a bit better because its likelihood function obeyed the $t$ distribution, which is more suitable for small ensemble sizes. For the variable $\eta$ without observation, AIb produced unstable results but was better than EIb and tXb. Compared with EIb, tXb produced more stable and better assimilation results. AIb produced poor assimilation results for $\omega$ and better but unstable results for $\eta$ in the sea-air coupled model. The conflict between these two variables was alleviated by EIb, which substantially improved the assimilation effect of $\omega$ and stabilized $\eta$ at the same time. Moreover, the new inflation scheme tXb improved the assimilation quality of these two variables again and reduced their RMSEs.

Figure 1b shows the experimental results for an ensemble size of 20. Although the sampling error was reduced, the same assimilation schemes with the inflation factors were still significantly better than the RMSEs of CTRL and SEO, which shows that covariance inflation significantly improves data assimilation quality. To compare the differences between the inflation schemes clearly, we did not compare these two schemes in subsequent experiments. When the ensemble size increased to 20, there was no longer a significant difference between EIb and tXb for $\omega$. The reason is that the larger ensemble size makes the $t$ distribution gradually approach the Gaussian distribution and produces a similar effect. For updated $\eta$ by the action of other variables only, consistent with the ensemble size of 5, tXb still had a better performance than EIb and was better than AIb here.

To intuitively compare the performance of different inflation methods for different ensemble sizes and to explore the implementation of the new inflation scheme between various sampling errors, we calculated the mean RMSEs of the last 5000 TUs for $x_2, \omega$ and $\eta$, and the results are displayed in Figure 2. The blue bar is the AIb scheme of A09, the orange bar is the EIb scheme of E18 and the yellow bar is the tXb prior inflation scheme. The results in Figure 1 show that the different inflation schemes have insignificant effects on the RMSE of $x_2$. However, the first subplot of Figure 2 shows that the new inflation scheme has some advantages over the other two for $x_2$ when the ensemble size is small, while the three schemes show comparable levels when the ensemble size exceeds 20. The advantage of the new scheme is more evident than the advantages of the others for $\omega$. When the sampling error was large, i.e., the ensemble size was 5, the effect of tXb improved by 48.6% relative to the classical AIb scheme. When the ensemble size was less than or equal to 20, tXb was better than EIb, which further indicated that the $t$ distribution of the likelihood function plays a major role for small samples. When the ensemble size was 5, EIb did not show any advantage for $\eta$. In contrast, except for a further reduction in RMSEs for $x_2$ and $\omega$, the effect of tXb improved by 45.9% for the unobserved variable $\eta$, reaching a similar level as AIb and showing a better effect than EIb. Throughout, tXb showed promising results when the ensemble size was small (Figure 2). tXb also offered comparable levels to EIb due to the gradual convergence of the $t$ distribution and the Gaussian distribution at larger ensemble sizes. This result indicates that tXb has better results than the other two prior inflation methods for most cases in the imperfect model, and the larger the sampling error, the more pronounced the effect.

In the simple sea-air coupled model, the variables $x_1, x_2, x_3$ and $\omega$ provided observations on general characteristics similar to those by most models. The variable $\eta$, which changed only under the influence of other variables, had more unique features and reflected the characteristics of the unobserved variables to some extent. So, the results of $x_2$ and $\omega$ showed that 20 ensemble members are enough to significantly reduce the sampling error in the imperfect simple sea-air coupled model. Due to the short integration time of the model, there is not enough capacity to respond to the changes due to the ensemble size, so an excessive ensemble size does not always give better results.

Also in this model, the computation times of different inflation schemes with different ensemble sizes were compared, as shown in Table 1. Due to the unstable computer power, the following values are the average results of three times of experiments. The time taken for the three schemes is close when using the same ensemble size, while the computation time will be significantly higher when the ensemble size increases. Therefore, using a small ensemble size can save more time cost.

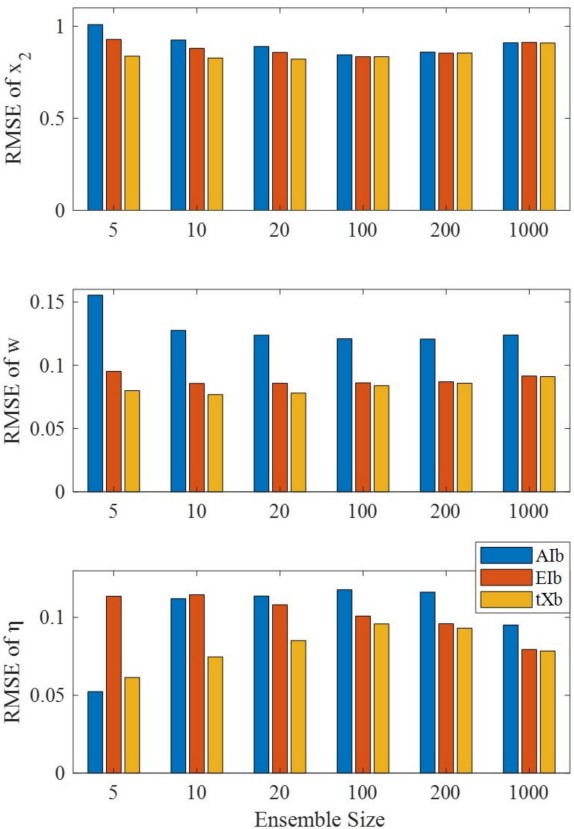

**Figure 2.** RMSEs of different ensemble sizes in the imperfect model with AIb (blue bar), EIb (orange bar) and tXb (yellow bar) for variable $x_2$, $\omega$, $\eta$. It is the time mean of the RMSEs in the last 5000 TUs.

**Table 1.** Comparison of calculation time for different inflation schemes.

| Ensemble Size | AIb | EIb | tXb |
|:---:|:---:|:---:|:---:|
| 5 | 1.6875 s | 1.9531 s | 1.9218 s |
| 20 | 5.2031 s | 5.4218 s | 5.4531 s |
| 100 | 23.4531 s | 23.6875 s | 23.9218 s |

- Posterior inflation scheme

The posterior inflation results for the three schemes are shown in Figure 3, where the ensemble size is 5 for (a) and 20 for (b). Similar to the results of the prior inflation schemes in Figure 1, the difference in the inflation schemes had no pronounced effect for $x_2$. For $\omega$, tXa was better than the other two schemes, especially AIa. For $\eta$, AIa showed better results when the ensemble size was 5, but tXa exhibited lower RMSEs when the ensemble size increased. In any case, tXa showed better results than EIa, indicating that the new inflation scheme is superior to the enhanced inflation schemes of E18 and E19 in some aspects.

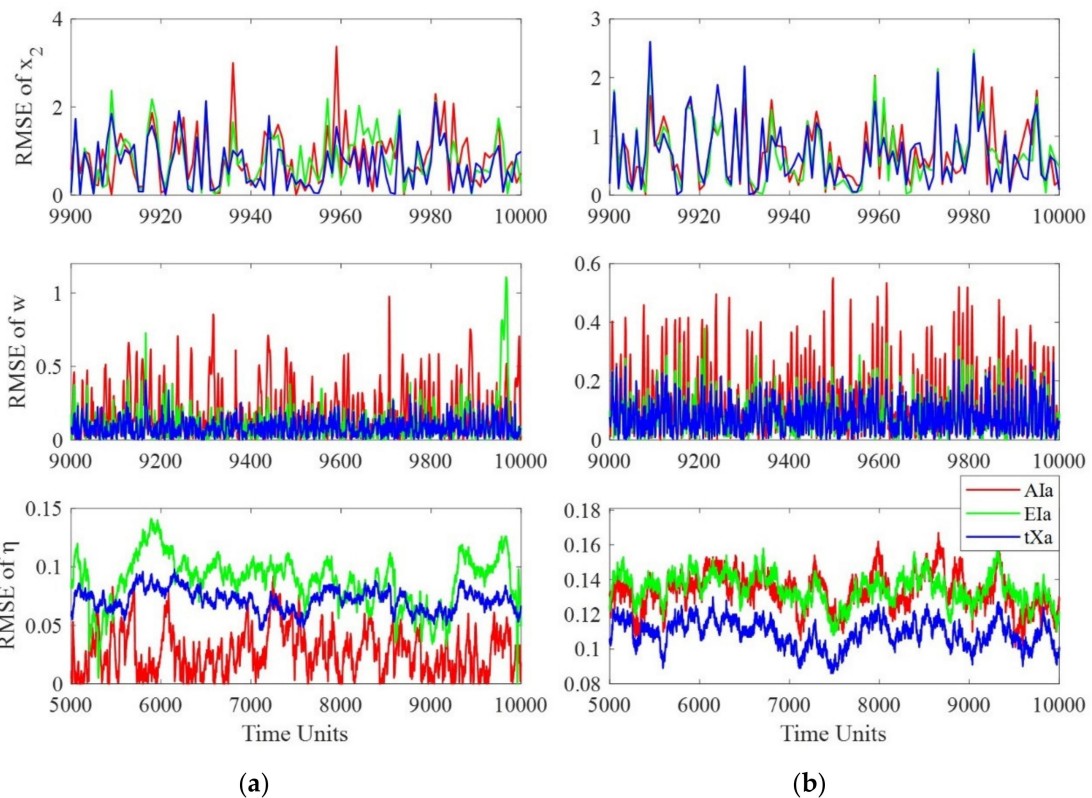

**Figure 3.** Time series of RMSEs of $x_2$ in the last 100 TUs, $\omega$ in the last 1000 TUs and $\eta$ in 10,000 TUs in the imperfect model with AIa (red line), EIa (green line) and tXa (blue line), with an ensemble size of 5 for (**a**) and 20 for (**b**).

The results of the prior and posterior inflation schemes of E19 and our schemes are compared in the same figure. We selected the variable $\eta$ with more stable RMSE results for comparison. The RMSEs of $\eta$ in the last 5000 TUs are shown in Figure 4, along with a high sampling error with an ensemble size of 5 for (a) and a low sampling error with an ensemble size of 20 for (b). The magenta line is the enhanced prior inflation scheme EIb, the red line is the new prior inflation tXb, the green line is the enhanced posterior inflation scheme EIa and the blue line is the new posterior inflation tXa. Irrespective of whether it is the prior or posterior inflation scheme, the result shows that the new inflation method outperforms the enhanced inflation scheme when the ensemble size is small (Figure 4a). Furthermore, all the inflation schemes were more stable for $\eta$ when the ensemble size increased to 20 (Figure 4b). The prior inflation was better than the posterior inflation for each scheme with an imperfect model and a small sampling error, which is consistent with the conclusion of E19. Moreover, the enhanced prior inflation scheme EIb had the same effect as our posterior inflation scheme tXa, indicating that the new inflation scheme is better than the enhanced scheme overall.

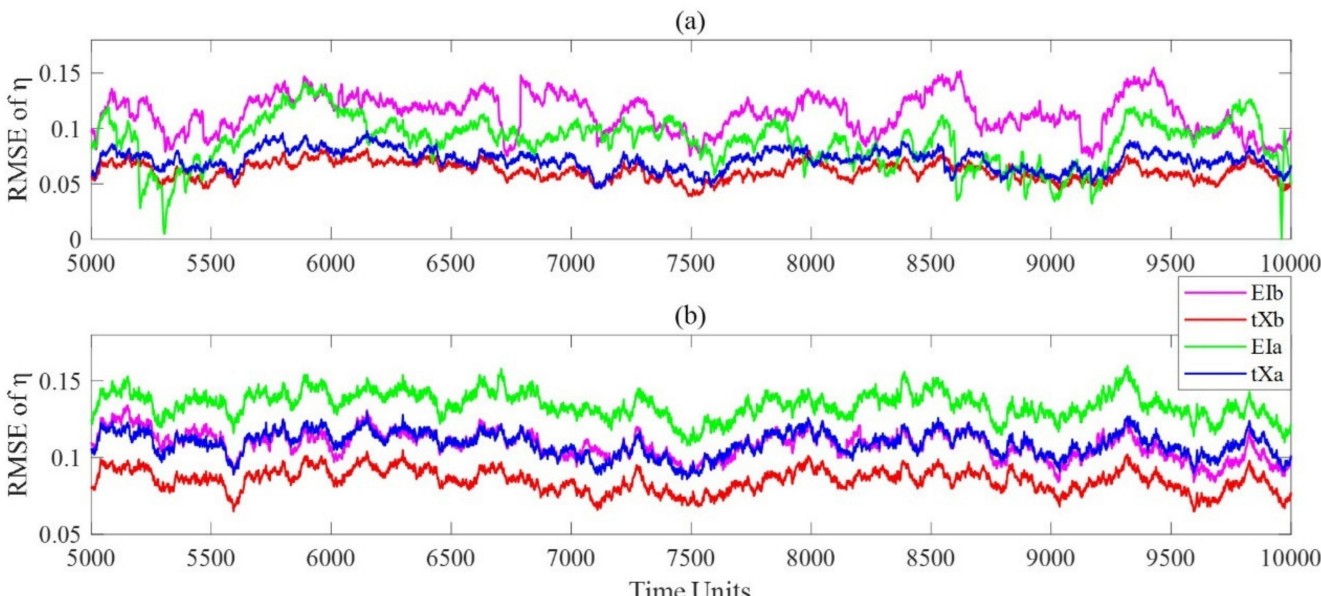

**Figure 4.** Time series of RMSEs of $\eta$ in the last 5000 TUs in the imperfect model with EIb (magenta line), tXb (red line), EIa (green line) and tXa (blue line), with an ensemble size of 5 for (**a**) and 20 for (**b**).

Perfect Model

In the assimilation framework with different inflation schemes, the unbiased model was integrated using the leapfrog scheme. As an example, the time series of RMSEs of $\eta$ obtained by using three prior inflation schemes are shown in Figure 5. The red line indicates the AIb scheme, the green line indicates the EIb scheme and the blue line indicates the tXb scheme. The results of an ensemble size of 5 are shown in Figure 5a and of 20 in Figure 5b. The RMSEs of $\eta$ in the perfect model showed a significant reduction compared to those in the imperfect model, and they were often close to 0. When the ensemble size was 5, the RMSEs of tXb were stable at a lower level, but the other two schemes increased much more suddenly at some moments and showed unstable results. When the sampling error reduced, the results of the three inflation schemes improved to some extent, but the new prior inflation scheme was still more stable for $\eta$.

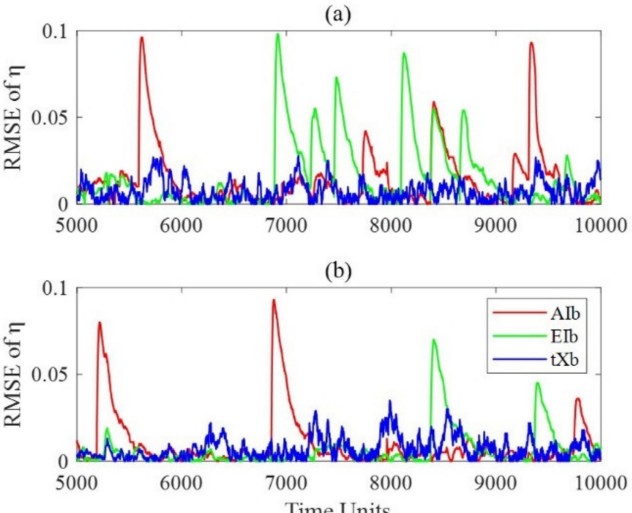

**Figure 5.** Time series of RMSEs of $\eta$ in the last 5000 TUs in the perfect model with AIb (red line), EIb (green line) and tXb (blue line), with an ensemble size of 5 for (**a**) and 20 for (**b**).

To better understand the influence of different sampling errors on the assimilation effect of the variable $\eta$ in the perfect model, the mean RMSEs of the variable $\eta$ in the last 5000 TUs with different inflation schemes are shown in Figure 6. tXb showed better or comparable levels compared with the other schemes in the perfect model regardless of the sampling error. However, similar to the result in the imperfect model, the special variable $\eta$ did not show the familiar regularity, but the RMSE decreased when the ensemble size was 100. The RMSE reached a low level because of the smaller sampling error and no model error, and the difference between the schemes was minimal. Such a slight difference is likely to occur by chance, and even different random noises may change.

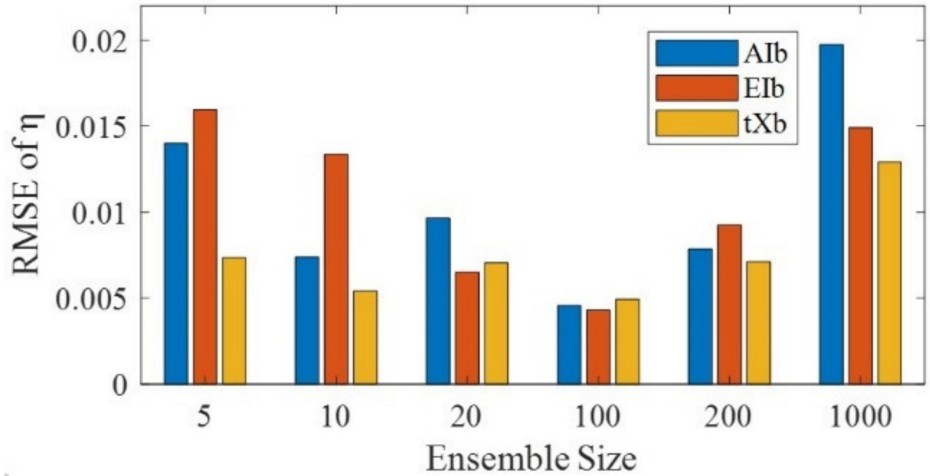

**Figure 6.** Mean RMSE of $\eta$ in the last 5000 TUs of different ensemble sizes in the perfect model with AIb (blue bar), EIb (orange bar) and tXb (yellow bar).

### 3.3.2. The Inflation Effect

To clearly show the advantage of adaptive covariance inflation, we compared the mean RMSE in the last 5000 TUs of tXb (blue) with that of SEO (orange) for different ensemble sizes in the imperfect model (Figure 7a) and the perfect model (Figure 7b). When the model error was large, tXb showed a significant advantage and a large ensemble size for SEO was still challenging to reach an equivalent effect. The RMSE of SEO significantly decreased without a model error, but tXb still performed better at high sampling errors. When the ensemble size increased and the sampling error gradually lowered, SEO had a similar effect to tXb, but it did not exist in the actual model. In the perfect model, the effect of tXb at an ensemble size of 5 was the same as that of SEO at 100 for $x_2$ and the effect of tXb at 5 was the same as that of SEO at 10 for $\omega$. The above results show that the scheme with the adaptive covariance inflation can effectively reduce the ensemble size, decrease the cost and speed up the computation.

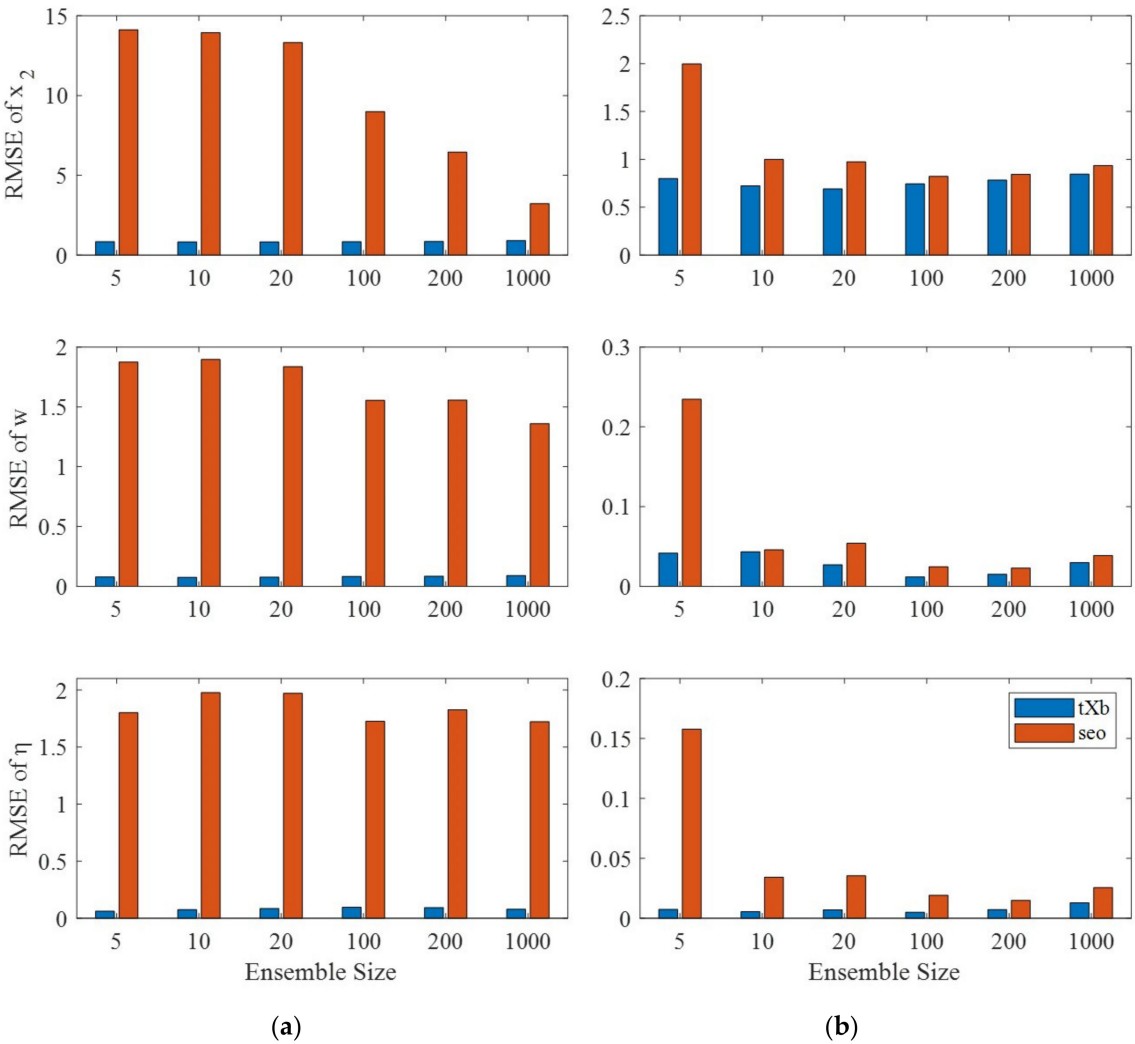

**Figure 7.** Mean RMSE in the last 5000 TUs of $x_2, \omega$ and $\eta$ for tXb (blue bar) and SEO (orange bar) with different ensemble sizes, using the imperfect model for (**a**) and the perfect model for (**b**).

## 4. MOCBM Experiments

### 4.1. The Model

After verifying the effect of the new adaptive inflation scheme by 5VCCM experiments, we also conducted experiments using another sea-air coupled model with a better physical basis [41]. The North Atlantic Meridional Overturning Circulation Box Model (MOCBM) [42,43] is a low-order model of the North Atlantic climate system consisting of an atmospheric model and an oceanic thermohaline circulation model. The former adds high- and low-latitude temperature variables to the three tropospheric variables in the low-order atmospheric circulation model proposed by Lorenz [44,45], which is different from the Lorenz63 convective model. The latter is a three-box ocean thermohaline circulation model, including the subtropical upper ocean, the subpolar upper ocean and the deep ocean. It evolved from the original two-box model [46], providing a basic understanding of the dynamics of the thermohaline circulation. In addition to the diffusion of temperature and salinity between each box, the upper ocean also exchanges energy with the atmosphere.

The two models are coupled through some variables and coefficients of the upper ocean and atmosphere, and the governing equations are:

$$
\begin{aligned}
\dot{X} &= -\left(Y^2 + Z^2\right) - aX + aF \\
\dot{Y} &= XY - bXZ - Y + G \\
\dot{Z} &= XZ + bXY - Z
\end{aligned}
\tag{30}
$$

where the dots above the variables denote the derivatives of the variables concerning time. $X$ denotes the zonal wind and $Y$ and $Z$ denote the amplitudes of cosine and sine phases of the large-scale eddies, respectively. $F$ denotes the diabatic heating contrasts between the low- and high-latitude ocean and $G$ represents the varying zonal heating zonal difference between land and ocean, both directly related to the upper-ocean temperature. The other terms and some of the meanings in the following equation are not described in detail here, and a detailed explanation can be found in the work of Tardif et al. [43]

The evolutionary governing equations for temperature and salinity for the three boxes are as follows:

$$
\begin{aligned}
V_1 \dot{T}_1 &= \tfrac{1}{2}q(T_2 - T_3) + K_T(T_{A1} - T_1) - K_Z(T_1 - T_3) \\
V_2 \dot{T}_2 &= \tfrac{1}{2}q(T_3 - T_1) + K_T(T_{A2} - T_2) - K_Z(T_2 - T_3) \\
V_3 \dot{T}_3 &= \tfrac{1}{2}q(T_1 - T_2) + K_Z(T_1 - T_3) + K_Z(T_2 - T_3) \\
V_1 \dot{S}_1 &= \tfrac{1}{2}q(S_2 - S_3) - K_Z(S_1 - S_3) - Q_S \\
V_2 \dot{S}_2 &= \tfrac{1}{2}q(S_3 - S_1) - K_Z(S_2 - S_3) + Q_S \\
V_3 \dot{S}_3 &= \tfrac{1}{2}q(S_1 - S_2) + K_Z(S_1 - S_3) + K_Z(S_2 - S_3)
\end{aligned}
\tag{31}
$$

where $T$ and $S$ denote the temperature and salinity in the ocean, respectively; $V$ denotes the volume of each box; and subscripts 1, 2 and 3 denote the high-latitude box, the low-latitude box and the deep-ocean box, respectively. $T_{A1}$ is the high-latitude air temperature, which is correlated with $X$, and $T_{A2}$ is the low-latitude air temperature, which is a constant 25 °C/298.15 K. $Q_S$ is the volume-averaged equivalent salt flux, which is linearly related to the eddy energy $(Y^2 + Z^2)$ [47]. The meridional overturning circulation (MOC) $q$ has a positive value in the thermal circulation [43] and presents a negative value in the reverse salt circulation, which is obtained from the temperature and salinity of the upper ocean as follows:

$$
q = \mu[\alpha(T_2 - T_1) - \beta(S_2 - S_1)]
\tag{32}
$$

where $\alpha$ is the thermal expansion coefficient of seawater, $\beta$ is the salt expansion coefficient and $\mu$ is the proportionality constant. The unit of $q$ is $Sv$, with $1\ Sv = 10^6 m^3 s^{-1}$. Other parameters are no longer listed for explanation, and the standard values of all parameters in MOCBM are set as $(a,\ b, F_0, F_1, F_2, G_0, G_1, G_2, V_1, V_2, V_3,\ \mu,\ \alpha,\ \beta, K_T, K_Z, T_{A2},\ \gamma, c_1,\ c_2,\ T_0)$ $= 0.25,\ 4.00,\ 6.65,\ 2.0,\ 47.9, -3.6,\ 1.24,\ 3.81,\ 0.832 \times 10^{16}\ m^3,\ 2.592 \times 10^{16}\ m^3,\ 10.30 \times 10^{16}$ $m^3,\ 4.0 \times 10^{10}\ m^3 s^{-1},\ 9.622 \times 10^{-5}\ K^{-1},\ 7.755 \times 10^{-4}\ psu^{-1},\ 3.5 \times 10^6\ m^3 s^{-1},\ 5.276 \times 10^5$ $m^3 s^{-1},\ 298.15\ K,\ 0.06364,\ 0.72 \times 10^6\ m^3 s^{-1},\ 0.015 \times 10^6\ m^3 s^{-1},\ 298.15\ K)$.

### 4.2. The Build-Biased Model

The next assimilation experiment required establishing an imperfect model. Given that this is achieved in the 5VCCM using different difference schemes, the MOCBM shows biased models using incorrect physical parameters. Since $q$ is directly related to the ocean state, we performed a sensitivity analysis of the physical parameters in the ocean and selected the most sensitive parameter to add bias to the experiment.

We used $q$ to test the sensitivity of parameters, following Zhao et al. [41]. The tested parameters were formed into an ensemble of 20 by adding Gaussian white noise with a standard deviation of 10% of its standard value, while the other parameters retained their standard values. The results were integrated freely for 250 years with the same initial field, and the last 200 years were taken to calculate the time-averaged spread. The sensitivity

percentage was obtained from the ratio of the sensitivity of a single parameter to the sum of the sensitivities of all parameters, as shown in Figure 8.

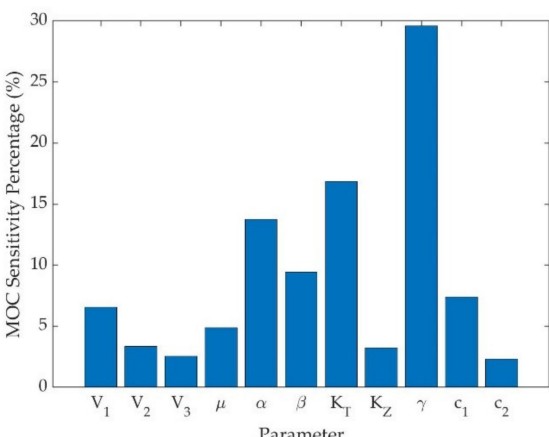

**Figure 8.** Sensitivity percentage of the MOC in parameter space. The spread of $q$ is obtained by the time average of the last 200 years.

Figure 8 gives the sensitivity percentages of all 11 physical parameters in the oceanic part of the model. Moreover, the MOC is most sensitive to the parameter $\gamma$, i.e., a change in $\gamma$ causes the most different values of $q$. Therefore, we added a 20% deviation to $\gamma$ to form a biased model with the wrong parameter, taking a standard value of 0.06364 for $\gamma$ in the perfect model and 0.076368 in the imperfect model.

*4.3. Experimental Design*

After comparing the similarities and differences between the new and other inflation schemes in the 5VCCM experiment, the main purpose of the MOCBM experiment was to verify the feasibility and effect of the new inflation scheme in this coupled model. The parameters in Section 4.1 and the values of the initial state in this section are from reference [48]. The perfect model uses the standard values of all parameters, and the imperfect model modifies the value of $\gamma$. We assumed that the only source of model error in this experiment is the incorrect physical parameters. Both perfect and imperfect models use a fourth-order Runge-Kutta time difference scheme with a time step of 3 h. Starting with the initial state field $(X, Y, Z, T_1, T_2, T_3, S_1, S_2, S_3) = (1.7, 0.0, 0.0, 288.15K, 298.15\ K, 283.15\ K, 34.21875\ psu, 35.0\ psu, 34.6\ psu)$, the model runs 2920 steps per year. Since the MOC has long timescale variables, it runs for 5000 years in this paper. The time series obtained by the perfect model is the true state. Following the feature of the existing observing system, observations are generated for only atmospheric and upper-ocean variables. In this study, the standard deviation of the atmospheric variables $X$, $Y$ and $Z$ were 0.1; $T_1$ and $T_2$ were 0.5 $K$; $S_1$ and $S_2$ were 0.1 $psu$; and the Gaussian white noise corresponding to the standard deviation was added to the true value with the observation frequency of 1 year to obtain the observation field of the model.

The MOCBM experiment also used the EAKF method for data assimilation, with 20 initial ensembles generated by adding white noise with standard deviation to the atmospheric variables $X$, $Y$ and $Z$ in the initial field. Two experiments were set up for comparison using imperfect models: the control (CTRL) experiment with free integration and the state estimation only (SEO) experiment.

A change in the ensemble size had little effect on the assimilation results in this model, showing that the model is not sensitive to the size of the sampling error, so this experiment selected an ensemble size of 20 for the investigation. In the perfect model experiment, the assimilation effect of SEO was excellent and the impact of adding the inflation factor was not apparent. The ideal model does not exist in practice; thus, the experiment of the perfect model was not conducted.

### 4.4. Result Analysis

In the imperfect model, comparison and assimilation experiments were performed with the same initial ensemble of the above parameter values and states. The average of 20 costumes was taken at each step and compared with the actual values to obtain the state time series, as shown in Figure 9.

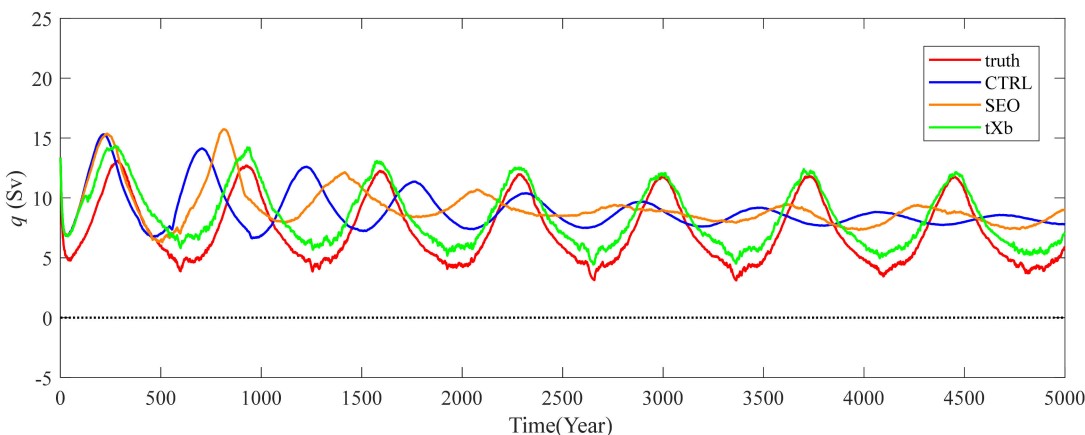

**Figure 9.** Time series of state means of the MOC *q*. CTRL (blue line), SEO (orange line) and tXb (green line) are compared with the true values (red line) for 5000 years running in the imperfect model, and the black dashed line indicating that *q* equals 0 is the dividing line between the two equilibrium states.

Figure 9 shows the 5000-year state time series of *q* from different comparison experiments. All the values were positive, which indicates that all the 5000 model years have heat-driven circulation, i.e., the ocean flows from the sea surface to the poles, sinks at high latitudes, returns from the deep ocean to the equator and upwells to the upper ocean at low latitudes [43]. All the experiments were compared using their ensemble mean values, and the results of both the CTRL (blue line) and the SEO (orange line) differed significantly from the actual state (red line) due to significant model errors. They did not even match the period of change. The tXb scheme with the new adaptive prior inflation factor (green line) fit the true value better and had the same period of variation, which benefits from the "observed" restrictions on the states and the adjustment of the various inflation factors. Therefore, the new inflation scheme is also applicable to the MOCBM with more obvious physical characteristics, and the adaptive inflation method described in this paper for the sea-air coupled model is feasible.

### 5. Discussion and Conclusions

A new adaptive covariance inflation algorithm was designed in this paper, including prior and posterior schemes. Based on Bayesian theory, the prior pdf of inflation obeyed the $\chi^{-2}$ distribution and the likelihood function obeyed the *t* distribution suitable for small samples. At the same time, the enhancement of the innovation statistic *d* presented in E18 was used, i.e., a correction was added to the inflation factor and the new adaptive prior inflation tXb was finally obtained. Based on the prior inflation scheme, the decorrelation in E19 was used for the posterior inflation scheme. In the first experiment, the adaptive prior inflation scheme in A09 was first used for the posterior ensemble and was compared with our proposed new scheme and the enhanced method in E18 in the framework of a simple sea-air coupled model. The parameters in the model were not changed, and the model errors only originated from different time difference schemes in the first experiment. The true state field was obtained by the leapfrog scheme, adding Gaussian white noise to generate observations, while the same technique was used for the perfect model integration. Furthermore, the RK4 scheme was used for the imperfect model. A series of experimental results were obtained by changing the ensemble size in the prior or posterior inflation

scheme. The second experiment added a model error using incorrect parameters to verify the new inflation scheme's feasibility for other coupled models.

The results show that the new prior inflation tXb has good performance in terms of some parameters compared with the other two schemes in the imperfect model. When the ensemble size was large, the effect of tXb was close to that of EIb because the $t$ distribution tended to be Gaussian. For the posterior inflation scheme, the effect of tXa was still better in most cases. Whether it was the prior or the posterior inflation, the new inflation scheme outperformed the enhanced scheme when the ensemble size was small and had no significant difference for a larger ensemble size.

In conclusion, the new inflation scheme in the imperfect model performs well for a small ensemble size, and it may be more suitable for high-dimensional, large-scale models. In the perfect model (although rare in reality), the new inflation scheme shows more stable results than the other two schemes. However, the results of the experiments are more affected by random errors due to minor sampling errors. Compared to SEO, tXb shows better results, especially in the imperfect model and tXb with a small ensemble size achieves the same effect as SEO with a large ensemble size in the perfect model.

The new inflation scheme also has some positive effects on the simple coupled model. However, there are still some limitations in this study, and the possible future research directions as follows.

1. The method has not been used in a real model, so further testing of the inflation scheme in real atmospheric and ocean models is needed.

2. We have assumed that the state variables are consistent with the observations, i.e., the projection operator $h$ is a unitary matrix, so further verification is needed when $h$ is not a unitary matrix or the observations are not perfect.

3. Due to computer performance limitations, we only performed a small number of iterations for the two simple models. In fact, a longer computation is necessary to reflect the physical processes of the models more clearly.

**Author Contributions:** Conceptualization, A.S. and L.Z.; methodology, S.Z. and X.Z.; software, Z.L. and C.L.; formal analysis, A.S. and L.Z.; writing—original draft preparation, A.S.; writing—review and editing, S.Z. and A.Z.; funding acquisition, L.Z. and S.Z. All authors have read and agreed to the published version of the manuscript.

**Funding:** This research was funded by the National Key R&D Program of China (2018YFC1407402, 2017YFC1404100, 2017YFC1404104 and 2018YFC1407401), the National Natural Science Foundation of China (Grant Nos. 41830964, 41775100, and 11801402), Shandong Province's "Taishan" Scientist Project (ts201712017), and the Qingdao "Creative and Initiative" frontier Scientist Program (19-3-2-7-zhc).

**Data Availability Statement:** Due to privacy-related restrictions, the data presented in this study are not publicly available but are available on request from the corresponding author.

**Acknowledgments:** The authors thank anonymous reviewers and editors for their thorough examination and comments, which were useful for improving the manuscript.

**Conflicts of Interest:** The authors declare no conflict of interest.

## Appendix A

In addition to the explanation of abbreviations in the text, all abbreviations and definitions are listed in alphabetical order in Table A1 to read more conveniently.

**Table A1.** Abbreviations and definitions used in this paper.

| Abbreviation | Definition |
|---|---|
| AIa | Adaptive posterior inflation |
| AIb | Adaptive prior inflation |
| CTRL | Control experiment |
| CGCM | Coupled general circulation model |
| EAKF | Ensemble adjustment Kalman filter |
| EIa | Enhanced posterior inflation |
| EIb | Enhanced prior inflation |
| EnKF | Ensemble Kalman filter |
| ETKF | Ensemble transform Kalman filter |
| MOCBM | North Atlantic meridional overturning circulation box model |
| RTPP | Relaxation-to-prior-perturbation |
| RTPS | Relaxation-to-prior-spread |
| SEO | State estimation only experiment |
| tXa | $t - \chi^{-2}$ posterior inflation |
| tXb | $t - \chi^{-2}$ prior inflation |
| 5VCCM | Five-variable coupled climate model |

## Appendix B

We approximated the Taylor expansion of the likelihood function as A09, leaving only the linear terms of a lower order:

$$p(d|\lambda) = \underbrace{p(d|\lambda_b)}_{\bar{l}} + \underbrace{\left.\frac{\partial p(d|\lambda)}{\partial \lambda}\right|_{\lambda_b}(\lambda - \lambda_b)}_{l'} + R(\lambda) \tag{A1}$$

where the last term is the residual term. $\bar{l}$ and $l'$ are represented as:

$$\bar{l} = \frac{\Gamma\left(\frac{v+M}{2}\right)}{(\pi v)^{\frac{M}{2}}\Gamma\left(\frac{v}{2}\right)}\left|\frac{v-2}{v}\bar{\theta}^2\right|^{-\frac{1}{2}}\left[1 + \frac{d^2}{(v-2)\bar{\theta}^2}\right]^{-\frac{v+M}{2}} \tag{A2}$$

$$l' = \bar{l}\left[\frac{(v+M-1)d^2 - (v-2)\bar{\theta}^2}{(v-2)\bar{\theta}^2 + d^2}\right]\bar{\theta}^{-1}\left.\frac{\partial \theta}{\partial \lambda}\right|_{\lambda_b} \tag{A3}$$

where:

$$\bar{\theta} = \sqrt{\sigma_o^2 + \left(\lambda_o - \frac{1}{N}\right)\hat{\sigma_b}^2} \tag{A4}$$

$$\left.\frac{\partial \theta}{\partial \lambda}\right|_{\lambda_b} = \frac{1}{2}\hat{\sigma_b}^2\gamma\left(1 - \gamma + \gamma\sqrt{\lambda_b}\right)\bar{\theta}^{-1}\lambda_b^{-\frac{1}{2}} \tag{A5}$$

However, in the actual calculation of Equation (A2), because of the larger degrees of freedom, the gamma function can produce results of huge orders of magnitude, out of the calculation range of a typical computer, even though the ratio of the two gamma functions is not large. To simplify the calculation of $\bar{l}$, the properties of the gamma function need to be used:

$$\frac{\Gamma\left(\frac{v+M}{2}\right)}{\Gamma\left(\frac{v}{2}\right)} = \frac{\Gamma\left(\frac{v+M-1}{2} + \frac{1}{2}\right)}{\Gamma\left(\frac{v-1}{2} + \frac{1}{2}\right)} = \frac{\frac{(v+M-2)!!}{2^{\frac{v+M-1}{2}}}\sqrt{\pi}}{\frac{(v-2)!!}{2^{\frac{v-1}{2}}}\sqrt{\pi}} = \frac{(v+M-2)!!}{(v-2)!!}2^{-\frac{M}{2}} \tag{A6}$$

To avoid calculating the double factorial with large magnitude in the numerator or denominator, the product of the quotient should be calculated by expanding the numerator

and denominator instead of calculating the quotient of double factorials. Specifically, when $M$ is an even number:

$$\frac{(v + M - 2)!!}{(v - 2)!!} = v \cdot (v + 2) \cdot \cdots \cdot (v + M - 2) \tag{A7}$$

When $M$ is odd and $v$ is odd:

$$\frac{(v + M - 2)!!}{(v - 2)!!} = \frac{2}{1} \cdot \frac{4}{3} \cdot \frac{6}{5} \cdot \cdots \cdot \frac{v - 1}{v - 2} \cdot (v + 1) \cdot \cdots \cdot (v + M - 2) \tag{A8}$$

When $M$ is odd and $v$ is even:

$$\frac{(v + M - 2)!!}{(v - 2)!!} = \frac{1}{2} \cdot \frac{3}{4} \cdot \frac{5}{6} \cdot \cdots \cdot \frac{v - 3}{v - 2} \cdot (v - 1) \cdot (v + 1) \cdot \cdots \cdot (v + M - 2) \tag{A9}$$

The above method avoids direct calculation but uses another way to obtain the value of the gamma function without exceeding the range of computer calculations. Thus, the value of $\bar{l}$ can be obtained for a large ensemble size.

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
