# Peer review of "A New Scheme of Adaptive Covariance Inflation for Ensemble Filtering Data Assimilation"

_jmse, doi:10.3390/jmse9101054_

Round 1

Reviewer 1 Report

The article presents a new inflation procedure to be applied in the context of atmospheric and oceanic global circulation models for a better estimation of state variables using Ensemble adaptive Kalman Filter EAKF processes.  

The procedure presented consists of a modification of the inflation factor at each assimilation step assuming that it is a statistical parameter described by an appropriate statistical distribution. The new procedure is part of a vein of literature, which sees a recent renewal of interest, proposing a Bayesian approach for calculating the posterior probability of the inflation factor. The a priori probability distribution, for the inflation factor, is assumed the inverse chi 2 and the t distribution is adopted as the likelihood distribution.

The new procedure is tested using atmospheric and oceanic global circulation models through two numerical experiments aimed respectively at highlighting the performance in case of errors in the numerical discretization and in the model parameters.

The results are evaluated in comparison with the schemes adopted in [28] and [29] also by testing the effects of the size of the Kalman filter ensemble.

The work is well planned and represents a very useful innovation in the scientific sector in which it is inserted. In my opinion, it can be improved by answering / discussing the considerations listed below.

1 - The reference to the procedure adopted in [14] - referred as A09 - is exhaustive. Not the same is the explanation of the method adopted in [29] - referred as E19. Being very concise, the reader has no way of evaluating the innovations without having recourse to the consultation of [29].

2 - In the overall evaluation of comparison between the tested procedures, there are no considerations about the computation times that they involve; in particular, when an increase in the size of the ensemble makes the performance of the procedures comparable, it is of interest to evaluate the savings in computing time the new procedure produces.

3 - The numerical experiments, assuming that the state variables were consistent with the observations, did not consider the need to evaluate the observations. It is appreciable that the Authors include some considerations on the application of the proposed method even in the case of treatment of the observations and, even more interestingly, in the evaluation of the model parameters.

Reviewer 2 Report

This study proposes a new scheme of adaptive covariance inflation for ensemble filtering data assimilation. I think the paper fits well the scope of the journal and addresses an important subject. However, a number of revisions are required before the paper can be considered for publication. There are some weak points that have to be strengthened. Below please find more specific comments:

*The abstract seems to be adequate. I suggest adding a sentence or two in the abstract to highlight the outcomes of this work and contributions to the state-of-the-art.

*The manuscript contains quite a few abbreviations. I suggest creating a table or an appendix that clearly defines all the abbreviations used.

*There are quite a few equations in the manuscript and many of them are not supported by the relevant references. Please include the supporting references for the adopted equations where appropriate to justify their selection.

*Please consider moving some of the heavy equations in the appendix. Otherwise, some readers may get confused.

*Please elaborate a bit more regarding the input data used in the experiments. Supporting references would be helpful.

*The conclusions section should expand on limitations of this study and future research needs. I suggest listing the bullet points.
